# Dynamic Pricing Models and Negotiating Agents: Developments in Management Accounting

**Edgard Bruno Cornacchione [1], Luciane Reginato [1], Joshua Onome Imoniana [1,\*] and Marcelo Souza [2]**

[1] School of Economics, Management and Accounting, University of Sao Paulo, Prof Luciano Gualberto Ave 908, Sao Paulo 05508-010, Brazil

[2] FIPECAFI (A Fundação Instituto de Pesquisas Contábeis, Atuariais e Financeiras), R. Maestro Cardim 1170, Bela Vista, Sao Paulo 01323-001, Brazil

**\*** Correspondence: josh.imoniana@usp.br; Tel.: +55-11-99244-9055

**Abstract:** Linking decision systems, negotiating agents, management accounting, and computational accounting, this paper aims at exploring dynamic pricing strategies of a synthetic business-to-consumer online operation and a comparative analysis of evolving strategy-specific pricing optimization. Five price models based on market, utility, or demand information (three single and two combined), merging online and offline data, are explored over a seven-day period and with twenty selected products. A total of 17,529 website visits and 538 agent negotiations are studied (94,607 main data points) using a Python solution, with model simulation parameters and assumptions described. Findings show the combined market-utility-demand performance of dynamic pricing to be superior as an input to the negotiating agent. Contributions are threefold, pointing to (a) management accounting practice and research (dynamic pricing), (b) science and research strategy (method), and (c) accounting education (skill set).

**Keywords:** dynamic pricing; negotiating agents; management accounting; computational accounting approach; data analytics; revenue management; buyer behavior; business-to-consumer

## 1. Introduction

Trade has always been a vital part of societies, with numerous advantages for all parties. Accounting has also played a role in this evolution, making sense of transactions and enabling parties to explore new alternatives to improve business performance. More recently, with ever-increasing technological capability, providers and consumers (to name a few involved parties) have been exposed to the online market, business-to-consumer connections, and a plethora of data to support business decisions. Therefore, new opportunities are presented on a daily basis for managers and accountants, especially for those more actively interested in the interplay between technology and business.

Pricing decisions, in light of the evolution of organizational strategies and goal-setting environment, have morphed as well. Typical pricing approaches, rooted in specialized literature from the 50s and 60s, gave room to more alternatives taking advantage of a plethora of relevant data and technical models, including experimenting and comparing their effectiveness. With the advent of the internet, endless improvement of the business-to-consumer space, streamlined transactions, and improved knowledge about the consumer, the provider, and competitors required executives to improve their pricing strategies and tactics. Many are examples in this century of e-commerce players (Streitfeld 2016; Mohammed 2017) implementing some sort of pricing innovation. If handling the typical human pricing decision to algorithms or robots is still a challenge for many players, neglecting to accept advances and variants of how competitors are dealing with this issue may represent an even more threatening option. Not to mention the potential to be brought by negotiating agents in the realm of smart contracts. Studies about smart contracts have revealed themselves in the accounting area. Chow et al. (2021) explore the use of

smart implementation technology for detailed principles that smart contracts be used in reconnaissance scenarios with generally accepted accounting principles (GAAPs), opening a space generally for a detailed understanding of studies in the accounting field.

Management accounting has focused on and perfected the pricing problem, primarily from a cost-based, shared-efforts, and market-oriented perspective. However, even with all these debates, specialized management and management accounting literature is still shy in terms of focusing on the debate about new pricing solutions, including dynamic pricing. As an example, in February 2021, a search on the term "dynamic pricing" in the American Accounting Association digital library (http://meridian.allenpress.com, accessed on 10 February 2021) yielded only three results (one being an out-of-context entry from a 1986 paper in the Accounting Historians Journals: " . . . Schmalenbach's dynamic price-level-adjustment model which was an extension of his dynamic balance sheet theory dating back to 1908"). The other two articles are: (a) Cost Management Research and (b) Reconciling full-cost and marginal-cost pricing. It is noteworthy that the first is loosely related to the topic, and the second proposes a formal model by adopting the concepts of dynamic pricing and full-cost pricing (FCP) in a monopolist context. Along similar lines, a similar search (same database) with the term "negotiating agents" yielded just one entry (out-of-context paper "Teaching Negotiation Skills within an Accounting Curriculum" from 2012 in Issues in Accounting Education: "I stress such risk issues as Russian, Venezuelan, and Chinese negotiation agents lacking clout in terms of their ability to bind corporations . . . "). Lastly, a similar search with the term "smart contracts" (same database) yielded 47 results, mainly devoted to the realms of systems and auditing.

With these three searches (dynamic pricing, negotiating agents, and smart contracts), we may gauge the coverage by specialized literature in the accounting field. On the other hand, in the field of technology, a search with the term "dynamic pricing" in the IEEE Xplore database (http://ieeexplore.ieee.org, accessed on 10 February 2021) yielded 214 journal articles. An analysis of the entries shows that the field of technology has engaged in debating not only the architecture of the solution but also business elements related to it, such as cost, demand, price, optimization, etc. Two immediate reactions to this: (a) accounting may need to address this allegedly curricular gap in terms of technology and anticipation of business model dynamics, and (b) accounting and technology could find ways of improving both academic and market collaboration.

Thomke (2020) explored the concept of experimenting with organizations and discussed the power of strategic business experiments. Hampel et al. (2020) focus on the role of experimentation to explore new ventures, having start-ups in mind, but addressing the benefit in terms of identifying market opportunities and ways of exploiting them. Such approaches are well suited for pricing, and we can find many organizations looking forward to reshaping the market, especially when goods and services are more commoditized, and price sensitivity is amplified.

Evidence of such organizational efforts on the topic, a search into the USPTO patent database, updated on 12 February 2021, with the Cooperative Patent Classification (CPC), Section G (Physics), Class G06 (Computing, Calculating, Counting), Subclass G06Q (Data processing systems or methods, specially adapted for administrative, commercial, financial, managerial, supervisory or forecasting purposes; systems or methods specially adapted for administrative, commercial, financial, managerial, supervisory or forecasting purposes, not otherwise provided for), yielded a total of 160,689 patents. A narrower search with the term "dynamic pricing" yielded 1219 patents. A combined search with "dynamic pricing" in the Description/Specification field and "G06Q" as the CPC class reached a total of 665 patents, with 55% of all patents granted in the past seven years (since 2014, 364 of all patents). Some examples are: (a) "Dynamic pricing method and apparatus for communication systems" (Motorola, US5303297, 1994); (b) "Dynamic pricing system" (Google, US7133848B2, 2006); (c) "Pricing model system and method" (Avanous, US7379922B2, 2008); and (d) "Dynamic pricing systems and methods" (Walmart, US10896433B2, 2021); and (e) "System and method for determining real-time optimal item pricing" (Visa, US10885537B2, 2021).

In short, 665 patents (55% in the last seven years) versus 3 and 214 academic papers, respectively, in accounting and technology repositories. These numbers represent the movement around the issue.

The goal in this study is to critically evaluate the parameters and outcomes of five dynamic pricing models within a controlled simulation environment. The study is driven by the following research question: To what extent can market-, utility-, or competition-based dynamic pricing be integrated into negotiating agents in business-to-consumer endeavors? This may add to the management accounting literature bringing perspectives to organizational strategy and pricing technology.

Contributions of this study are threefold, pointing to (a) management accounting practice and research, (b) science and research strategy (method), and (c) accounting education. Such contributions are presented in detail along with the conclusion of this paper.

## 2. Literature Review

Pricing is a key strategic piece for management, intrinsically related to management accounting and, also a business challenge as "it is simultaneously affected by cost and demand conditions which are not parallel and are difficult to align as an efficient decision supporting the strategic objectives of the firm" (Laitinen 2011, p. 311). Many areas of expertise, such as economics, accounting, business, law, psychology, and engineering, just to mention a few, devote substantial energy to exploring and explaining the phenomenon. Such a variety of backgrounds in investigating the topic does not always align, as is the case between accounting and economics explored by Lucas (2003). In addition, organizations, supported by scientific evidence or not, invest energy and time as well to outpace competitors and favor consumers while benefiting the bottom line. In other words, this has been and still is a very relevant topic in business.

As part of the revenue management realm, dynamic pricing has been gaining prominence in the past decades (Narahari et al. 2005) mainly due to affordances in business, technology, human expertise, and market data (Maglaras and Meissner 2006), or as stated by Aviv and Vulcano, "in the past, firms did not possess the capability to plan, execute, and take advantage of dynamic pricing strategies" (Aviv and Vulcano 2012, p. 2). Exploring data from more than 1700 B2B business leaders, a recent report on the topic states that "85% of management teams believe their pricing decisions need improvement, and only 15% have effective tools and dashboards to set and monitor prices" (MacArthur et al. 2020, p. 64). A similar report, based on data from 2005 global business leaders and data from projects, stresses the potential of pricing optimization to have a 2% to 5% financial impact in less than nine months (Chugani et al. 2020). A recent study suggests a global market value impact of up to USD 500 billion due to artificial intelligence pricing and promotion initiatives (Chui et al. 2018)

Driven by the effects of technological advances on the management accountant role, Andreassen (2020) conducted a study with evidence from a Nordic insurance company and claimed that "the increased sophistication of pricing models and use of various data sources and types contributes to specialization in the pricing process ... [becoming] more refined and frequently updated to reflect changes in the underlying data" (p. 222). Combining massive amounts of data, pricing technology absorbs more business strategy as inputs and becomes "more sophisticated in how they use data and computer power to model the pricing elements and customer behavior" (p. 223).

Dynamic pricing, in the COVID-19 pandemic situation, also can help promote better overall experience and value, as is the case of a study focusing on vehicle routing and home delivery (Straussa et al. 2020).

As simply put by Calvano et al. (2020), we live on the verge of algorithms supplanting human decision-makers when it gets to pricing goods and services. Not only that, but we are observing "collusive behaviors of autonomous pricing algorithms" not actually as a planned action but as a result of their own learning process (Calvano et al. 2020, p. 3). An example of the availability of artificial intelligence pricing solutions is the Interactive Pricing Analytics

Pre-Configured Solution, an Azure cloud application (e.g., Cortana Intelligence Gallery) able to suggest prices for wholesale and retail, using elasticity and transaction records.

Another element that arises from this debate is that the consumer sense of fairness and the perception of the provider (goods or services) may be at a crossroads when it comes to discriminant or dynamic pricing. The tourism industry is paying attention to this, in light of equity theory (Adams 1965) and debating the reasons for a consumer to pay more than another, as stated by Khandeparkar et al. (2020) on the practice of dual pricing, highlighting formal research is still shy of exploring the perception of price unfairness. Focusing on the unfairness of dynamic pricing, Santos et al. (2019) explored the advances in network companies, such as Uber and its surge pricing, and consumer behavioral strategies. Regarding people's behavior and the delegation of prices, Wamsler et al. (2022) demonstrated in their study that dynamic pricing affects customer perceptions and influences their decisions.

Another aspect of this debate is the legal dimension, often referencing the U.S. Code (Robinson-Patman Act of 1914), passed during the U.S. Great Depression: "it shall be unlawful for any person engaged in commerce . . . to discriminate in price between different purchasers of commodities of like grade and quality" (15 U.S.C. § 13). The debate is intense, and many forms of support are found, giving space to charge different prices to different customers. Beyond the legal aspect, authors focus on the unethical or exploitative abuse behind price discrimination (Mohammed 2017; OECD 2018; Streitfeld 2016).

A more traditional approach to pricing in the field of business, economics, and accounting always brings costs to the equation, with the seminal work of Hall and Hitch (1939) bringing direct and indirect costs into the economics literature. Or as stated by Farm (2020), "it is an empirical fact that cost-plus pricing is a common pricing procedure in a market economy" (p. 61). Particularly nowadays, to enhance market equilibrium in view of social well-being towards a circular economy. Based on environmental, economic, and social dimensions, which aims to ensure sustainable development at each step of product creation, transformation, and conversion by creating a closed-loop economy (Nikonorova et al. 2020).

Management accounting literature deals with a range of organizational and market conditions, from price takers to price makers, often referring to the Amoroso-Robinson rule: assuming that "only information about marginal cost and price elasticity of demand are relevant" (Laitinen 2011, p. 323). Typically, pricing strategies reflect one of three main assumptions (Bitran and Caldentey 2003; Cunningham and Hornby 1993; Hinterhuber 2008; Schuster et al. 2021; Sunarni and Ambarriani 2019): (a) cost-based pricing (using data from cost accounting), (b) competition-based pricing (using price data from competitors), or (c) customer value-based pricing (using the utility or value of goods and services for customers). Relevant literature is signaling this potential prominence of the value or utility aspect in pricing decisions when considering the more current market and economic conditions (Kienzler and Kowalkowski 2017), whereas studies still bring empirical evidence of a more traditional cost-based approach (Correa et al. 2016; Sunarni and Ambarriani 2019).

Beyond the technical aspect, price decisions must deal with a key behavioral element: customer willingness to pay the price from a dynamic pricing strategy. This may be more or less relevant depending on the industry, as evidence (Anderson and Xie 2016; Bandalouski et al. 2018) shows that customers accept distinct pricing in the hospitality industry (e.g., hotels, airlines, entertainment).

Thus, the selected literature synthesizes the current status of relevant discussions on key variables to this study, such as the strategic relevance of pricing, its debate across distinct areas of expertise, new technology and market data influence on pricing, the financial impact of pricing optimization, the impact of artificial intelligence on pricing, and the recent sophistication of pricing models, including the prominence of algorithms. In addition to this, the selected literature stresses the risks long associated with having different prices for the same products or services. Based on such aspects, we were able to

build the research question driving this study, which seeks to explore the feasibility (and intensity) of embedding dynamic pricing models into digital negotiating agents.

We also observe an intense movement in academia and organizations related to pricing optimization and revenue management, based on the selected literature, and enough room to explore these elements as developments in management accounting.

## 3. Method

This is a study based on the computational accounting approach (Wall and Leitner 2020) proposing a simulation model focusing on external (market) and internal (organization) information and price-settling agreement-seeking negotiating agents (Jonker et al. 2012). It relies on the simulation research method, more specifically the agent-based model method, which has a bottom-up (part to whole) approach, where agents " . . . interact with each other and their environment, resulting in emergent outcomes at the macroscale" (Heckbert et al. 2010, p. 40). The model is based on specific assumptions and parameters (see Appendices A and B) and reflects an online business-to-consumer organization with its product line and the goal of experimenting with pricing technologies. The research hypothesis of the study is: Selected dynamic price models fed into the negotiating agents yield significantly different revenue performance. Or in other words, there is a superior price model (and consequently derived price points) for negotiation purposes. This research hypothesis is directly linked to the research question of this study and aims to explore the feasibility and intensity of embedding dynamic pricing into autonomous digital negotiating agents.

### 3.1. Sweet Indurgency

"Sweet Indurgency," a Sweets Network Company (SNC) and our model organization, has its online presence with a clear and dynamic website in line with the premium sets (product line) that are sold to the public in distinct regions. Due to the observed growing competition, management started to study dynamic pricing strategies and now decided to start a pricing experiment with 20 products for seven days. In order to accomplish that, they refined the current inventory, supply, cost, and price information for 20 products, overall presenting about 25 percent of the contribution margin. The initiative involved, on the first day, to start collecting more precise data on website visits and customer intention (based on an algorithm influenced by customers' interactions with the website). Following this, days 2 to 7 (six days) were actually active for the dynamic pricing experiment.

Systematically a system to monitor competitors (not necessarily direct but with similar products) was implemented, and from day 1 on, data was collected informing a market price pressure indicator for each of the 20 products (the "market" dimension). In addition, a system to keep track of the website visits was implemented and generated metrics as a daily heat map of the products (the "demand" dimension). In addition, the intention (purchase orientation) of each visitor was tracked by another system generating an intention indicator (the "utility" dimension).

### 3.2. Dynamic Pricing Models and Negotiating Agents

A dynamic pricing solution was created to present the prospective customer with an attractive price for both parties. In addition to the existing price list, the experiment was supposed to test five options and inform a decision on which option would be selected to fulfill the dynamic pricing model. The five options are (a) market-based (following 100% of the price pressure from competitors), (b) utility-based (influenced by intention as a proxy for customer utility), (c) demand-based (directed by the heat map generated by website visits), (d) utility-demand (a combination of the last two options), and (e) market-utility-demand (a combination of the last option with the first one).

The dynamic pricing model under scrutiny and its connected negotiating agent are able to generate a customer-specific price for each product of interest based on these five approaches. The customer is presented with the specific price for the transaction and is able to decide about the purchase. The customer can negotiate the price in rounds with the

autonomous negotiating agent, and that will happen automatically when not confirming the transaction. When leaving the cart, the dynamic pricing negotiating agent will offer its second-best price (from the perspective of the company) in the following negotiation round.

The default strategy of the negotiating agent is to offer the highest price between the utility-demand price and the market-utility-demand price. In case of follow-up negotiation, the other price (of these two) is offered. The autonomous negotiating agent will seek to improve the overall rate of accept/reject decisions by customers. The overall customer behavior model will feed back to the dynamic pricing system as a whole and on a daily basis (a current limitation of this version of the negotiating agent).

Parameters of the synthetic company allow it to obtain an average of 2500 visits to its website per day, with about 4 percent of highly interested prospects potentially reaching the dynamic pricing agent. Thus, overall revenues are estimated to be about USD 2.5 million per year.

*3.3. Technology Protocol and Code*

In terms of the computational environment, the micro-organizational world to model the "Sweet Indurgency" pricing situation was created using a readily-accessible and scalable cloud solution: Python 3.7.10 in Google Colab® (http://colab.research.google.com) with two high-performance Intel Xeon CPU with 2 GHz (Family 6, Model 85, Skylake) and memory allocation of 12.72 Gigabytes. A typical execution of the simulation of the model, with all parameters, uses about 0.92 Gigabyte of memory and takes, on average, about 100 s to perform all tasks. The code has four main parts and relies on these imported Python libraries: (a) numpy 1.19.5, (b) matplotlib 3.2.2, (c) seaborn 0.11.1, (d) pandas 1.1.5, (e) scipy 1.4.1, (f) time, and (g) statistics. The first part is the library import section, followed by the second part with the functions: iSupply, iCompetitors, prospects, heatMap, askPrice, pricePressure, newVisit, and negotiate. The third part is the simulation routine with the timer, simulation parameters, initial setup of the data frames (supply, competitors, visits, customers, and deals), and the simulation core (rounds of daily simulations). The fourth and final part consists of model auditing, statistical analysis, reporting routine, and timer close-up (for efficiency metrics).

**4. Data Analysis**

In order to perform the data analysis, we added the model parameters to the simulation engine (see Appendix A), and then we loaded the price-cost-supply data (see Appendix B) to it. All these elements (parameters and initial data) were carefully planned considering the "Sweet Indurgency" synthetic company and market scenario (see Section 3, "Method"), in line with the study about dynamic pricing in the electronic business developed by Narahari et al. (2005). The results are directly dependent on the engine itself and the parameters.

The core of the simulation (without auditing, statistics, reporting, and visualization part) took 101.09 s to perform, handling a total of 94,607 main data points. Results are now presented and discussed. With 20 products in this seven-day period process, a total of 17,529 website visits was recorded, and a total of 538 negotiations (3.1 percent) with 481 deals actually made, yielding a total revenue of USD 58,684 in the period (USD 2.8 million, annualized). Thus, the negotiating agent could reach an agreement with engaged prospects 89.4 percent of the time. Such results can be linked back to the study of Chugani et al. (2020) in terms of the financial impact of pricing optimization, as well as to Calvano et al. (2020), stressing the prominence of algorithms in pricing.

For the entire simulation, Table 1 shows the number of visits, deals reached and proportion, and overall customer buying intention.

**Table 1.** Characteristics of the simulation (all products).

| Day | Visits | Deals | Deals (%) | Intention (%) |
|-----|--------|-------|-----------|---------------|
| 1 | 2455 | 99 | 4.03 | 45.05 |
| 2 | 2503 | 89 | 3.56 | 44.36 |
| 3 | 2518 | 80 | 3.18 | 45.01 |
| 4 | 2454 | 89 | 3.63 | 44.91 |
| 5 | 2532 | 98 | 3.87 | 44.75 |
| 6 | 2546 | 83 | 3.26 | 45.02 |
| 7 | 2521 | 99 | 3.93 | 44.94 |

The dynamic pricing engine in this model operates with six prices: (a) asking price (cost-plus-margin), (b) market price (market-based), (c) utility price (utility-based), (d) demand price (demand-based price), (e) utility-demand price, and (f) market-utility-demand price. Such a strategical approach to pricing is aligned with Laitinen (2011) and with the main management accounting pricing strategies (Bitran and Caldentey 2003; Cunningham and Hornby 1993; Hinterhuber 2008; Schuster et al. 2021; Sunarni and Ambarriani 2019). Analysis involving prices in this model is presented next.

Beyond the typical asking price (cost-plus-margin), the market price in this model was set based on competition price pressure of similar quality lines of products offered in the market. Figure 1 shows a skewed distribution reflecting, for the simulation (all products and the entire period), an average competitor price pressure indicator of 49.6851 (in a 0–100 scale) and standard deviation of 2.4514, unveiling a negative pressure on prices for the analyzed period. Such negative pressure makes the market prices go down, affecting customer expectations and behavior and influencing the negotiating agent in this model. By considering this level of market information, the results are aligned with propositions by Maglaras and Meissner (2006) and Aviv and Vulcano (2012).

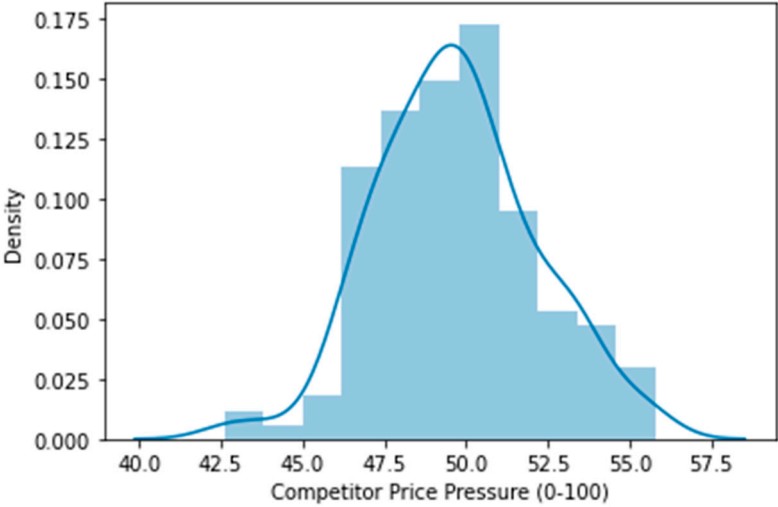

**Figure 1.** Overall competitor price pressure distribution.

From Figure 2, it is possible to observe that there is no simple (or linear) explanation for negotiation agreement based on price concepts. For instance, sometimes the market price (orange) is at the top (greater), sometimes at the bottom (lower), while the deal price (pink) tends to stay in the middle of other prices, very close to the market-utility-demand mix.

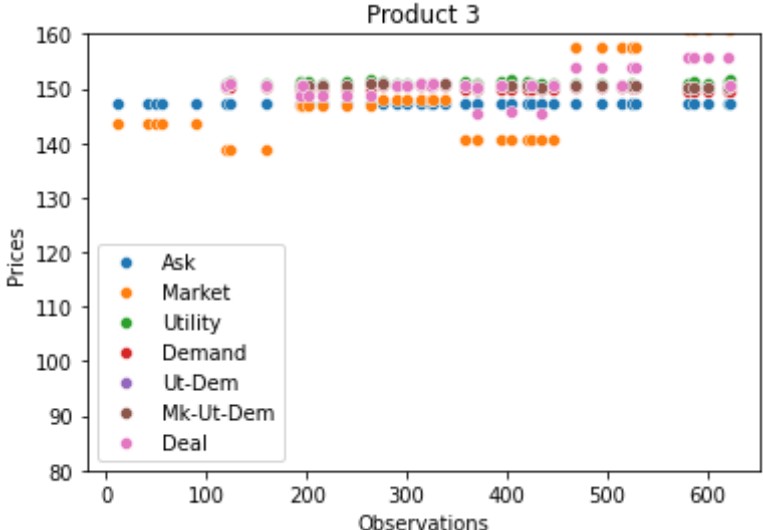

**Figure 2.** Price comparison for product 13.

Customer intention is key in this model, as it serves as a proxy for utility-based pricing, and Figure 3 brings relevant data on this. For the entire simulation, all products, and the entire period, the average customer intention was 44.8613 (on a 0–100 scale), and the standard deviation was 13.7905, showing some challenges for deals across the analyzed period.

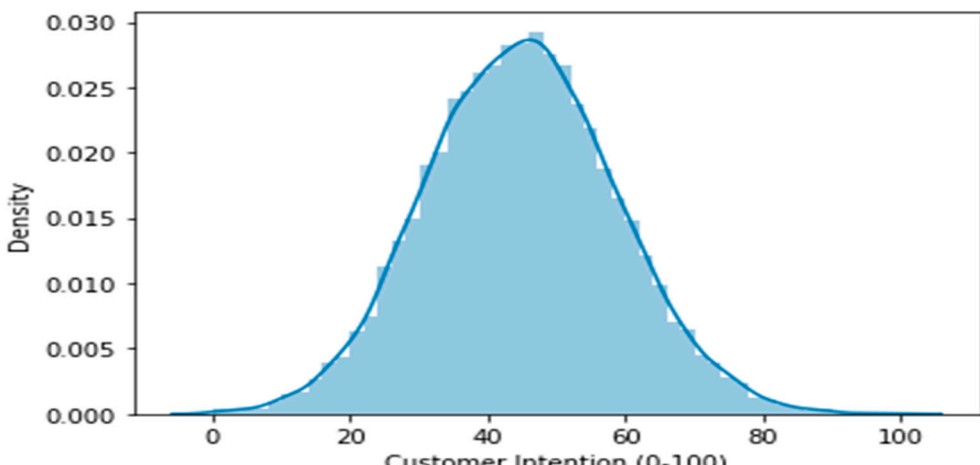

**Figure 3.** Overall customer intention distribution.

It is also possible to observe, in Figure 4, that phenomenon by comparing the density function of the deal price (blue) and the market price (orange) for product 14. Market prices are concentrated (higher level, approximately between USD 60 and USD 70), while deal prices fluctuate more (due to the dynamic pricing negotiating agent) at lower price levels.

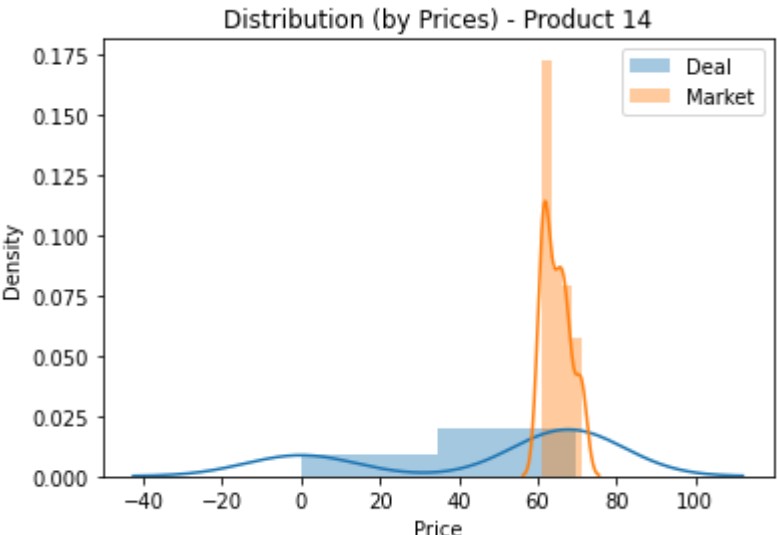

**Figure 4.** Distribution of deal and market prices of product 14.

In summary, despite the risks long associated with having different prices for similar products or services, as explored by Adams (1965), Khandeparkar et al. (2020), Santos et al. (2019), Wamsler et al. (2022) and the Robinson-Patman Act (1914), in this study, based on dynamic pricing models used by negotiating agents to reach deals with prospective customers, we observed results aligned with Anderson and Xie (2016), where customers are willing to negotiate and accept distinct prices.

### 4.1. Comparison of Means

We compared, with *t*-tests, all price models adopted by the dynamic pricing negotiating agent in this study with the deal price to discover which pricing models tend to act more closely to the mutual interests of customers and the negotiating agents. Results are presented next. When comparing deal prices with market-utility-demand (MUD) prices, no differences were found. With utility-demand (UD) prices, just for product 14, we found significant differences ($-2.0871$, $p = 0.039231$). The same was the case with (D) demand-prices; just for product 14, we found significant differences ($-2.1091$, $p = 0.037245$). A similar situation was found with utility-based (U) prices for product 14 ($-2.0672$, $p = 0.041108$). When comparing to market-based prices, all products showed significant differences, but products 2, 7, 9, 16, 18, and 20). In addition, when comparing with the asking price, all showed significant differences, but products 7, 9, 16, and 18). Thus, in this study, the combination of the market-utility-demand (MUD) price model represents the best approach for the negotiating agent to perform and reach more agreements with engaged customers.

### 4.2. ANOVA

For all price models adopted by the dynamic pricing negotiating agent in this study, we conducted an Analysis of Variance to check whether such prices were similar or not. Results are presented in Table 2. This is another way to look at the price discrimination models fed to the negotiating agent, which can be visualized in Figure 2 (with an example of variating prices).

**Table 2.** ANOVA Results.

| Product | ANOVA (F-Value) | *p*-Value |
|---|---|---|
| 1 | 2.5655 | 0.019159 * |
| 3 | 2.2256 | 0.040907 * |
| 4 | 4.6287 | 0.000165 * |
| 5 | 3.1088 | 0.008136 * |
| 6 | 2.7071 | 0.014911 * |
| 10 | 2.8575 | 0.012318 * |
| 11 | 2.4371 | 0.025856 * |
| 12 | 3.2266 | 0.004102 * |
| 14 | 4.6043 | 0.000158 * |
| 15 | 3.0463 | 0.006225 * |
| 17 | 4.1602 | 0.000648 * |

* All significant at the 0.05 alpha level.

As we can observe from Table 2, all products but 2, 7, 8, 9, 13, 16, 18, 19, and 20 (neutral product bias, see Appendix A) present significant variances of prices at 0.05 alpha level. Combining this with other analyses, we can assert the volatility of prices among the large set of products (twenty) with distinct characteristics and parameters and claim the superiority of the combined MUD price model in reaching deals with engaged customers.

## 5. Discussion

Initially, it is important to state that reaching the findings in this paper was largely driven by two major forces: (a) the absence of a mature discussion in the relevant literature bound to the element of dynamic pricing and computational advances embedded in negotiating agents and (b) the opportunity of having a functional agent-based model synthesizing the complexities of the simulated business-to-consumer online operation for the purposes of studying and comparing pricing models and strategies. Findings of this study back the final claim about integrating dynamic pricing models into digital negotiating agents and, in doing so, optimizing deals. In addition, findings show the superior performance of the combined pricing model (market-utility-demand) studied here.

In the study, with customer intention (see Figure 3) at the 44.86 level (on a 0–100 scale) and negative price pressure at the 49.68 level (on a 0–100 scale) from competitors, presenting a challenging environment for sales performance, the negotiating agent reached deals in 89.4% of the time, yielding an annualized total revenue of about USD 2.8 million). Such metrics suggest the relevance of digital negotiating agents powered by dynamic pricing models (in line with Calvano et al. 2020) and the need for developing the computational accounting area, according to Wall and Leitner (2020), both in academia and organizations, as it represents an edge to management, by adding value to the organization.

With such results, it is clearly possible to ignite the pricing discussion within management accounting studies situated within the strategic business experiments and fueled by advanced technology powerfully integrating massive amounts of data from cost, operations, market, utility, and demand, in light of Thomke (2020), Hampel et al. (2020) and Laitinen (2011). In fact, optimizing pricing decisions and proper revenue management is a real demand (pain point) from organizations, as stated by MacArthur et al. (2020) and Chugani et al. (2020), with (potentially) enormous financial impacts, as described in the study by Chui et al. (2018). The implications of such findings, in both academic and practical worlds, may lead to advanced technological business solutions, as the digital negotiating agent created and used in this study can accept (and learn) a myriad of models powered by a distinct set of parameters to better tackle the operations and market at hand. In a data-driven world, organizations now have been accumulating vast amounts of

data, ready to be transformed into actionable assets in the interest of optimized business performance, as suggested by the study of Andreassen (2020).

Another point to be highlighted while interpreting the results with the selected management accounting literature is the potential power of an integrated approach to a variety of pricing models long adopted by companies and studied by academics, as mentioned by Cunningham and Hornby (1993). In this study, we combined cost-, competition- and customer-driven pricing models (Bitran and Caldentey 2003), admitting that in certain market conditions (context), there may be a superior approach, such as market and demand (Kienzler and Kowalkowski 2017) or cost-based (Sunarni and Ambarriani 2019; Farm 2020). We did that supported by advanced technology and programming such flexibility into the digital negotiating agent so that it was granted autonomy to deal with a specific set of parameters and adjust the offer by exploring distinct pricing models on the fly, or exactly as the negotiation stages were developing.

Another aspect derived from the findings of this study is the volatility in price ranges (see Figure 2) across the twenty products present in the simulation and the comparison of deal prices: see Figure 4 stressing product 14, as well as the comparison of means and the ANOVA (Table 2) sections, both bearing statistically significant differences in said prices. Such price volatility and deal price distance represent, together, a powerful attribute of this dynamic pricing agent-based model.

Finally, the computational ability to access and evaluate, on the fly or as the negotiation evolves, large amounts of data and signals backed by a robust set of pricing models linked to a clear sales strategy fuel, as shown in this study, the digital negotiating agent, as envisioned by Jonker et al. (2012), in a way to improve organizational performance.

## 6. Conclusions

The pricing quest represents a relevant challenge for many organizations, not to say all, and the revised literature in this study shows an accelerated pace in certain areas of knowledge towards taking advantage of technology and experience to improve revenue management and pricing optimization, whereas contributing to the consumer experience. We explored dynamic pricing strategies of a synthetic business-to-consumer online operation (over a seven-day period) and developed a comparative analysis of evolving strategy-specific pricing optimization, by using (in addition to the traditional cost-plus-margin) five price models based on market, utility, or demand information (three single and two combined). Relying on a computational accounting approach we were able to design and develop a cloud software solution, using Python and a series of readily available programming libraries to explore the issue of dynamic pricing with the help of negotiating agents. After many rounds of exploration and improvements of the model and computing solution (mainly functions and parameters), we were able to analyze data (94,607 main data points) and get into the details of 438 deals. No other study of this nature was found in the management accounting literature, stressing its relevance.

Management accounting has a powerful set of concepts and tools to support a precise comprehension of business variables, their relationship, and their consequences. The current socio-economic scenario and expected transitions in organizations, notably in price-setting technology, require accounting practitioners and scholars to devote time and energy to the issue. Based on the outcomes from the repositories of accounting and technology studies and patents, we stress the need for management sciences, specifically management accounting, practitioners, and scholars, to act and have a stronger presence in this debate.

In this study, we were able to tackle the set research hypothesis ("Selected dynamic price models fed into the negotiating agents to yield significantly different revenue performance"), and we claim that there is a superior price model (and consequently derived price points) for negotiation purposes. Here, we found that a combination of market-, utility-, and demand-based price models fed into negotiating agents was capable of reaching higher revenue levels. No other model could reach similar levels of revenue. This result was supported by all developed analyses (including a comparison of means and ANOVA) in

this study, in a negative price pressure scenario (Figure 1), price volatility (Figure 2), and challenging customer intention (Figure 3).

Surprisingly, the market-based price model presented a weak performance in this study. It seems that for companies acting as price-takers (a very significant number, indeed, due to market conditions), a big portion of business may be hurt by relying on a mimetic approach to pricing strategies. This study shows that such companies could be better off if acting towards a slightly more sophisticated price model with specific pricing automation technology.

The answer to the set research question ("To what extent can market-, utility-, or competition-based dynamic pricing be integrated into negotiating agents in business-to-consumer endeavors?"), considering the claim just presented is to a large extent. We are now able to assert that dynamic pricing models can be integrated into business consumer situations, such as those considered in this study. The said integration, as discussed in this study, was reached by creating a digital negotiating agent powered with dynamic pricing models and embedding it into the digital selling process. Such an agent was able to generate a customer-specific price for each product of interest based on market-, utility, or competition-based dynamic pricing models, interacting and reaching deals with potential customers 89.4% of the time.

A direct interpretation of this is that pricing is still a complex issue and is influenced by many variables, and even with the adoption of technology to improve the negotiating process while minimizing rejections (in negotiations), there is a need for a cautious selection of concepts and models that make more sense to the business process at hand. This is aligned with recent studies in the field (Axtel and Farmer 2022; Recio et al. 2022). No one-size-fits-all solution is expected here. Even "over-the-counter" computational solutions require extreme care when setting up parameters and configuring the environment according to the culture and nature of the business.

*6.1. Contributions*

Contributions of this study are threefold, pointing to (a) management accounting practice and research (dynamic pricing), (b) science and research strategy (method), and (c) accounting education (skill set). Being able to explore buyer behavior and price-setting strategies with massive amounts of details (parameters and data), is critical when performing forward-looking analysis. This can be viewed as critical for policy-makers acting in different directions of the market (including government authorities, legislators, and market regulators). Policy-makers need to address the environment and conditions where dynamic pricing technology will perform. For instance, government authorities, respecting their jurisdiction, tradition, competence, and interests may benefit from anticipating a set of principles or rules to address legitimate concerns of the market players. In addition, legislators may have to formally address the societal debate related to the level of technological presence in the market. Market regulators, on the other hand, may have to explore the risks associated with managerial technology, especially in more automated negotiation interactions, to prevent unfair market practices.

In price-setting processes management accounting, both scholars and practitioners may be able to take advantage of dynamic pricing studies, as we observed, aligned with recent studies, plenty of room to advance the literature on the topic and organizational process related to dynamic pricing supported by advanced agent technology, including those potentially embedded in smart contracts. Secondly, the ability to navigate in the modeling dimension, with support from current strong computational affordances (e.g., hardware, databases, and programming languages, such as Python), to mimic and emulate relevant behaviors for management accounting is clearly a strength of the method and could become more present among scholars. Lastly, in the education realm, it becomes natural to have such skills, knowledge (and attitude) regarding both the topic (dynamic pricing) and the technology behind it more available to students in the field of management accounting, as we see an uneven acceleration across disciplines, leaving accountants with a

risk of not being able to cope with fast developments and arrangements in organizations with regards to a typical topic in accounting literature, such as pricing.

*6.2. Limitations*

Although this study highlighted many positive aspects of the use of models and simulations to help with the development of dynamic pricing in management accounting, we may also stress the limitations of the study that is naturally bound by a set of assumptions and parameters to feed the model and its computational simulation rounds. This is to say that conclusions could only be generalized if bound by a similar scenario and to an equivalent set of assumptions and parameters, as adopted here. Thus, the scenario and context (e.g., organization, product line, cost and supply setup, customer base, etc.) are limiting elements themselves. An important limitation of the current version of the dynamic pricing negotiating agent is related to how customer behavior is fed back into the system. The current agent does this on a daily basis and uses an overall customer behavior input per product. This could be improved by advancing the feedback system to a real-time process with an individual (returning customers) approach. Although the current agent is capable of supporting the claims in this study, this anticipated improvement may afford different conditions and consequent new claims about dynamic pricing based on fine-tuned customer behavior information, potentially achieving higher levels of accept/reject customer decisions. The model tends to mimic reality but with levels of fidelity only up to a certain point in the interest of computational efficiency and variable sensitivity. In other words, scholars may find it interesting to explore in more depth a specific parameter or set of variables having to change the scope and intensity of this particular model, potentially yielding distinct results and interpretations.

**Author Contributions:** Conceptualization, E.B.C. and L.R.; methodology, E.B.C.; software, J.O.I.; validation, E.B.C., L.R. and J.O.I.; formal analysis, E.B.C.; investigation, E.B.C.; resources, L.R.; data curation, J.O.I.; writing—original draft preparation, E.B.C.; writing—review and editing, J.O.I.; visualization, J.O.I.; supervision, L.R. and M.S.; project administration, M.S. All authors have read and agreed to the published version of the manuscript.

**Funding:** This research received no external funding.

**Institutional Review Board Statement:** Not applicable.

**Informed Consent Statement:** Not applicable.

**Data Availability Statement:** Not applicable.

**Conflicts of Interest:** The authors declare no conflict of interest.

## Appendix A

**Table A1.** Model Parameters.

| Parameters | Set Value |
|---|---|
| Simulation Days (first day is for historical purposes only) | 7 |
| Base Price Variation | 0.015 |
| Base Prospects (per day) | 2500 |
| Bias Seed (strong bias 0.15, weak bias −0.15) | −0.11 |
| Chance Returning Customer | 75 |
| Threshold Purchase | 70 |
| Pressure Bias | −0.001 |
| Offer 1 (chance of accepting 1st offer) | 0.60 |
| Offer 2 (chance of accepting follow-up offer) | 0.15 |
| Pressure Seed (randomized pool, set for a slightly negative biased pressure) | −0.009, −0.003, −0.003, −0.001, 0, 0, 0.001, 0.002 |
| Pressure Price | f(Pressure Seed, Pressure Bias) (normal distribution, M = 10, SD = 0.5) |
| Product bias (load factor = 1, neutral) | 2, 5, 6, 7, 8, 9, 10, 13, 16, 17, 18, 19, 20 |
| Product bias (load factor = 2, twice as much) | 1, 3, 4, 11 |
| Product bias (load factor = 3, three times) | 12, 14, 15 |
| Intention Seed (randomized pool, set for a slightly positive biased intention) | −0.07, −0.04, −0.01, 0, 0.02, 0.05, 0.08 (normal distribution, M = 10, SD = 3) |
| Heat Map (per product, per day) | f(Intention, Visits) |
| Spot Intention (per product, per day) | f(Customer, Intention) |
| Deal Seed (randomized) | (normal distribution, M = 5, SD = 2) |

## Appendix B

**Table A2.** Initial Setup of Product Attributes: Cost, Price, Supply.

| Product | Cost | Price | Supply |
|---|---|---|---|
| 1 | 88.24 | 118.1 | 100 |
| 2 | 42.75 | 57.25 | 70 |
| 3 | 110.41 | 147.3 | 150 |
| 4 | 136.78 | 182.72 | 180 |
| 5 | 16.21 | 21 | 40 |
| 6 | 77.69 | 104.13 | 100 |
| 7 | 59.71 | 79.1 | 55 |
| 8 | 44.99 | 59.8 | 78 |
| 9 | 123.12 | 165.12 | 120 |
| 10 | 125.52 | 167 | 110 |
| 11 | 55.85 | 75.75 | 80 |
| 12 | 136.82 | 182 | 200 |
| 13 | 35.38 | 47.11 | 470 |
| 14 | 49.54 | 66.1 | 660 |
| 15 | 143.65 | 191.99 | 500 |
| 16 | 64.76 | 86.64 | 100 |
| 17 | 51.23 | 69 | 150 |
| 18 | 78.72 | 105.12 | 200 |
| 19 | 33.12 | 46.2 | 300 |
| 20 | 76.85 | 107.8 | 100 |

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
