# Peer review of "Dynamic Pricing Models and Negotiating Agents: Developments in Management Accounting"

_admsci, doi:10.3390/admsci13020057_

Round 1

Reviewer 1 Report

This is a review of the paper “Dynamic Pricing Models and Negotiating Agents: Developments in Management Accounting” that was submitted to “Administrative Sciences”.

The aim of the paper is not very clear. It is different in the abstract and in the introduction.

From the abstract: “this paper aims at exploring dynamic pricing strategies of a synthetic business-to-consumer online operation and a comparative analysis of evolving strategy-specific pricing optimization”.

From the introduction: “The goal in this study is to critically evaluate the parameters and outcomes of five dynamic pricing models within a controlled simulation environment”.

I am not convinced that the research question was really answered.

From the introduction section:  To what extent can market-, utility-, or competition- based dynamic pricing be integrated into negotiating agents in business-to-consumer endeavors?

Is there a clear answer to this question?

The research hypothesis of the study is not based on a well-motivated research gap. The hypothesis seems obvious. Why would we claim a reverse?

From the method section: “Selected dynamic price models fed into the negotiating agents yield significantly different revenue performance”.

Isn’t it obvious?

The contribution is not detailed enough. It is vague.

From the abstract: “Contributions are threefold, pointing to (a) management accounting practice and research (dynamic pricing), (b) science and research strategy (method), and (c) accounting education (skill set).”

Authors deliberate further on these three specific contributions in the conclusions section, but it is not very convincing. Contribution should be as precise as possible.

The description of the method requires improvement, as it is hard to follow. Is it a real company? It would be beneficial to include an example of a deal.

Author Response

Dear Reviewer,

Attached, we have address your concerns.

Thamk you for your efforts,

Joshua

Reviewer 2 Report

Dear Authors, 

I found the topic interesting and valuable. But it is difficult to understand from the paper what the research is about. If based on the title, it could be management accounting and dynamic pricing. If you follow the introduction - the article could cover smart contracts, different pricing models and options, etc. So it is not clear what is the problem that the research is trying to focus on and how to contribute to solving it. Please clarify the problem and based on that the research question and research tasks.

Study design. It is not clear how the hypotheses were formed - a deeper analysis of the literature and an explanation of the formation of the hypotheses would be needed. For some reason, only one hypothesis is stated in the study design, but there seem to be more when it comes to interpreting the data.

Results. The results are very superficially interpreted. Some results are not interpreted at all (last table). The results need a deeper interpretation based on the literature.

Difficult to see the contribution of the current research' to the field of management accounting, as well as to the field of management and organization science. A more thorough research design (including hypothesis design), a clearer focus, a more comprehensive literature review, and a deeper analysis of the data could contribute to the field of pricing.

Author Response

Dear Reviewer,  

We appreciate your efforts in reviewing our manuscript.

Thank you,

Joshua

Reviewer 3 Report

The paper is highly interesting and very well written. Besides, it seems highly interesting to a large scope of readers. Notwithstanding, the scientific component of the paper must be improved to help users to understand its theoretical basis and also how the findings are useful and new. The literature review is relatively poor (and citations are not according to the journal requirements) and a research design is not presented to the readers. More specifically, I would suggest authors significantly improve the aspects below:

* The introduction is particularly focused on the gaps, but does not provide how this research intends to fill those gaps and its main contributions to the practice and to the literature;

* The literature is poor and does not discuss the variables used in a way that readers can understand the model to be tested;

* A scheme of the model and research design could be included in the literature or in the methods section to help readers to better understand the objectives behind this research;

* Discussion of results is not significantly provided by authors;  

* As a detail, the presentation of appendices can be improved.

Nonetheless, I think that the research has a significant interest and impact power if those elements are improved.

Author Response

Dear Reviewer,

We appreciate the efforts put in moulding our article.

Thank you,

Joshua

Round 2

Reviewer 1 Report

The paper is still hard to read (English is fine, it is rather the way Authors guide potential readers through the text). The literature review does not lead to the research question.

Authors write that “The goal in this study is to critically evaluate the parameters and outcomes of five dynamic pricing models within a controlled simulation environment. The study is driven by the following research question: To what extent can market-, utility-, or competition- based dynamic pricing be integrated into negotiating agents in business-to-consumer endeavors?”.

In my opinion Authors did not answer the research question. What is the short and precise answer to this research question? Could conclusions be generalized? Or maybe conclusions are specific to the presented study/company?

Conclusions are full of sentences that lack detailed, clear recommendations. For example Authors write “This can be viewed as critical for policy-makers acting in different directions of the market (including government authorities, legislators, and market regulators)”, but do not provide detailed guidelines.

Author Response

Reply in the attached file

Reviewer 2 Report

Thanks for developing the paper. Added sentences and paragraphs make the paper clearer. Relation to accounting and relevance to accounting well explained.

I would like to see a deeper analysis. The outcome of the research is presented, but the interpretation based on the theoretical framework of the outcome could be better.

Author Response

Reviewer's concern addressed

Reviewer 3 Report

Despite the changes made by authors and the argument that literature is still poor as regards the proposed topic, I still consider that the scientific soundness must be improved to allow authors to create 1) more robust hypotheses and a clear research model; 2) to improve the discussion of the results in light of the literature (a specific section would be highly appreciated). 

Author Response

Attached herewith, response to reviewer 3

Round 3

Reviewer 3 Report

Dear authors,

Thanks again for the improvements made. I recognize that it is highly difficult to improve the scientific soundness of a topic such as this. Notwithstanding, I would consider improving it by adding a specific section for discussion, using some of the considerations proposed within the conclusion, but providing an adequate link with the literature review. 

Please be also aware of the need to use the citation pattern proposed by this journal and to avoid some redundancies (for instance, "...considering the claim just presented, is to a large extent. We are 448 now able to assert that, to a large extent,..."

Author Response

The concerns of the Reviewer has been addressed.
